# Serum Neurofilaments and OCT Metrics Predict EDSS-Plus Score Progression in Early Relapse-Remitting Multiple Sclerosis

**DOI:** 10.3390/biomedicines11020606

**Published:** 2023-02-17

**Authors:** Vlad Eugen Tiu, Bogdan Ovidiu Popescu, Iulian Ion Enache, Cristina Tiu, Alina Popa Cherecheanu, Cristina Aura Panea

**Affiliations:** 1Department of Clinical Neurosciences—Department 6 (Neurology)—“Carol Davila” University of Medicine and Pharmacy, 050474 Bucharest, Romania; 2Neurology Department, Elias University Emergency Hospital, 011461 Bucharest, Romania; 3Neurology Department, Colentina Clinical Hospital, 020125 Bucharest, Romania; 4Neurology Department, University Emergency Hospital of Bucharest, 050098 Bucharest, Romania; 5Ophtalmology Department, University Emergency Hospital of Bucharest, 050098 Bucharest, Romania

**Keywords:** multiple sclerosis, EDSS plus, biomarkers, RRMS, neurofilaments

## Abstract

(1) Background: Early disability accrual in RRMS patients is frequent and is associated with worse long-term prognosis. Correctly identifying the patients that present a high risk of early disability progression is of utmost importance, and may be aided by the use of predictive biomarkers. (2) Methods: We performed a prospective cohort study that included newly diagnosed RRMS patients, with a minimum follow-up period of one year. Biomarker samples were collected at baseline, 3-, 6- and 12-month follow-ups. Disability progression was measured using the EDSS-plus score. (3) Results: A logistic regression model based on baseline and 6-month follow-up sNfL z-scores, RNFL and GCL-IPL thickness and BREMSO score was statistically significant, with χ2(4) = 19.542, *p* < 0.0001, R2 = 0.791. The model correctly classified 89.1% of cases, with a sensitivity of 80%, a specificity of 93.5%, a positive predictive value of 85.7% and a negative predictive value of 90.62%. (4) Conclusions: Serum biomarkers (adjusted sNfL z-scores at baseline and 6 months) combined with OCT metrics (RNFL and GCL-IPL layer thickness) and the clinical score BREMSO can accurately predict early disability progression using the EDSS-plus score for newly diagnosed RRMS patients.

## 1. Introduction

Multiple sclerosis (MS) is a chronic disease of the central nervous system. Demyelination, neuroinflammation and neurodegeneration are the core processes driving disability in MS, although there is much ongoing debate regarding how these interplay [1].

Relapse-remitting forms of MS (RRMS) benefit from a wide array of disease-modifying treatment (DMT) choices, and large registry studies have shown potential benefits for the early initiation of high-efficacy therapy options [2]. While this may also be a more cost-effective choice in the long run, emerging healthcare systems may struggle financially to enroll all newly diagnosed RRMS patients on high-efficacy DMTs [3]. Prognostic biomarkers may help to identify patients more likely to experience a more severe course of the disease in resource-limited settings.

Many such potential biomarkers have been investigated. Cerebrospinal fluid (CSF) beta-amyloid (Aβ), namely the Aβ42, might hold some value as a prognostic biomarker in MS [4,5], as previously confirmed by our group as well [6]. Serum and CSF neurofilament light chains (NfL) [7,8] are already well characterized biomarkers, and while CSF samples are usually analyzed as raw values, serum NfL (sNfL) may be adjusted using a digital tool that accounts for the patients’ weight and age in deriving a z-score or percentile for these samples (as described by Kuhle et al. [9]).

Imaging biomarkers have been long employed as biomarkers in MS. These include both classic and novel magnetic resonance imaging (MRI) techniques. Due to reasons highlighted in our previous review on this matter, we believe classic MRI metrics (such as T1 and T2 lesions, gadolinium-enhancing lesions (GdE) etc.) to be much more familiar to neurologists and radiologists worldwide at this point, and more relevant to daily practice as imaging biomarkers [10]. A proposal from the ECTRIMS workshop on aggressive MS stated that a worse prognosis may be expected from patients that show 20 or more T2/FLAIR hyperintense lesions on the baseline MRI or two or more GdE [11]. Guidelines regarding DMT efficacy assessment propose that three or more new T2/FLAIR hyperintense lesions or one or more new GdE lesions on follow-up MRIs may be considered to be a sign of inadequate treatment response [12].

Optical coherence tomography (OCT) provides an easy, direct view of the central nervous system by means of the optic nerve. The thickness of the axonal mesh that creates the retinal nerve fiber layer (RNFL) and the macular ganglion cell-inner plexiform layer (GCL-IPL) have been proven to be reliable prognosis biomarkers in MS [13,14,15,16,17].

Clinical prediction scores may also have value as predictive tools. One such example is the Bayesian risk estimate for MS at onset (BREMSO) score, designed to predict long-term prognosis for MS patients using clinical variables available following the first assessment [18]. Another such example is the RoAD score (risk of ambulatory disability). This score predicts the 10-year risk of ambulatory disability (EDSS = 6.0) using a mix of clinical and paraclinical data available after one year of follow-up [19].

Assessing acquired disability in the early stages of the disease may prove to be challenging. The expanded disability status scale (EDSS) is one of the most popular scales available for quantifying disability due to MS, but not without having some shortcomings as well [20,21,22,23]. EDSS-plus, a derived scale that incorporates metrics from the time to walk 25-feet test as well as the 9-hole peg test, was initially developed for assessing the progression of disability in secondary progressive MS (SPMS). However, data from clinical studies show a great sensibility even to the earliest accruals of disability, and we believe it may hold value as a tool for the early stages of RRMS as well [24,25].

We decided to perform a prospective cohort study in order to determine the correlations between early disability progression in newly diagnosed RRMS patients and the most commonly used predictive biomarkers available today.

## 2. Materials and Methods

Our cohort is comprised of 52 consecutive patients that were diagnosed with relapse-remitting MS (RRMS) from June 2020 to October 2021 in our center.

Patients were considered eligible if they had been recently diagnosed with RRMS. This was defined in our study as fulfilling the 2017 McDonald criteria [26] for RRMS at any point during the previous 6 months. Signed informed consent was mandatory for study participation, and subjects had to be at least 18 years old to be enrolled. Patients were excluded in case of ongoing pregnancy or any associated medical history that would interfere with the study protocol.

The study protocol involved a baseline evaluation comprising OCT analysis, contrast enhanced cerebral MRI scan and serum and CSF collection along with clinical evaluation. Follow-up visits were programmed at 3, 6 and 12 months after inclusion, and included serum sample collection and clinical evaluation. The final visit at one year also included follow-up contrast enhanced cerebral MRI. For the baseline serum and CSF samples, all patients were naïve to MS-related treatments.

OCT evaluation was performed within our hospital in the Ophthalmology Department using a CIRRUS™ HD-OCT 5000 machine. Data obtained from eyes that had been previously affected by optic neuritis were excluded from the final analysis.

All patients were followed-up for a minimum duration of 1-year.

### 2.1. Sample Processing Protocol

Serum samples were centrifuged after 40 min of clotting at 2000 rotations per minute (RPM) for 10 min. CSF samples were centrifuged immediately following lumbar puncture using the same protocol. Sample storage was conducted at −80 degrees Celsius.

Total CSF cell count, CSF glucose and CSF protein fractions were tested for all patients. Oligoclonal bands, as well as CSF beta amyloid (Aβ42 fraction) and neurofilament light chain (using the SIMOA assays method) were also analyzed.

### 2.2. Statistical Analysis

Approval for this study was obtained from the ethics committee of the University Emergency Hospital of Bucharest. Written consent was mandatory from all patients prior to inclusion in our study.

Statistical analysis was performed using SPSS 26.0 for Windows (SPSS Inc., Chicago, IL, USA) and Microsoft Excel 2019.

The Kolmogorov–Smirnov test and Shapiro–Wilk test were used to check for the normality distribution of data. The independent samples T test and Kruskal–Wallis test were used when comparing continuous variables. Categorical variables were compared using the chi-squared test and Fischer’s exact test.

The Cochran–Armitage test of trend was used to establish the correlation between dichotomous and ordinal variables. Binary logistic regression was performed to establish predictive models between the dichotomous variable of the EDSS-plus progressor state and measured biomarkers (either continuous or dichotomous variables).

## 3. Results

The study lot, consisting of 52 patients, was closely monitored for a minimum of one year. Of these, 37 patients were women (71.2%) and 15 were men (28.8%). The median age was 29 years at inclusion (18; 52). Table 1 shows the main characteristics of the patient lot.

An initial correction for outliers was performed for all of the considered biomarkers. Values above two standard deviations within the variable were defined as significant outliers, and we removed four entries from baseline CSF NfL, one from baseline sNFL, one from the one-year follow-up sNfl, one from the CSF Aβ42 and one from the OCT GCL-IPL from the final statistical analysis.

We wanted to explore which patients showed disability progression following RRMS diagnosis (early disability progressors). We decided to use the EDSS-plus score, considering as an EDSS-plus progressor any patient that showed either a 1.0 point increase in EDSS score after one-year follow-up or an increase of at least 20% in either the 9-hole peg test or the time to walk 25-feet test. At the one-year follow-up, 19 patients (36.5%) were characterized as EDSS-plus progressors, as opposed to 33 (63.5%) non-progressors.

We further analyzed possible confounders associated with EDSS-plus progressor status, as can be observed in Table 2.

As we can observe from Table 2, no statistically significant differences were found between the EDSS-plus progressor and non-progressor groups regarding possible confounders or known risk factors.

We further analyzed whether statistically significant differences were found between the EDSS-progressor status and the main biomarkers and clinical scores that were performed for this study. The main findings can be seen in Table 3.

Initial analysis using descriptive statistics showed a trend towards higher adjusted z-scores for sNfL in the EDSS-plus progressor group compared to the non-progressor group. Statistical significance was reached for the 6- and 12-month follow-ups. Final RoAD score and both RNFL and GCL-IPL thickness also showed statistically significant differences between the two groups.

We also analyzed whether there was a statistically significant impact of the DMT type on the sNfL samples collected after DMT initiation, namely the 3-, 6- and 12-month follow-up samples. No differences were found between the high-efficacy DMT group and moderate-efficacy group.

We further proceeded by performing a binomial logistic regression to see whether the analyzed biomarkers may be used to build a predictive algorithm for the progressor state of the EDSS-plus score.

Based on the results presented above and data from the literature, the final prediction model included the following biomarkers: baseline and six-month follow-up sNfL-adjusted z-scores, RNFL and GCL-IPL average thickness and the BREMSO score.

The linearity of the continuous variables with respect to the logit of the dependent variable was assessed via the Box–Tidwell procedure. A Bonferroni correction was applied. Based on this assessment, all continuous independent variables were found to be linearly related to the logit of the dependent variable. We further corrected for outliers, residuals and leverage points.

The logistic regression model predicting the likelihood of the EDSS-plus progressor state based on the mentioned biomarkers can be seen in Table 4.

BREMSO: Bayesian risk estimate for MS at onset; GCL-IPL: ganglion cell-inner plexiform layer; RNFL: retinal nerve fiber layer; sNfL: serum neurofilaments.

The area under the ROC curve was 0.968 (95% CI, 0.926 to 1.0), with an excellent level of discrimination according to Hosmer et al., as can be seen in Figure 1.

The logistic regression model was statistically significant: χ2(4) = 19.542, *p* < 0.0001. The model explained 79.1% (Nagelkerke R2) of the variance in the EDSS-plus progressor state and correctly classified 89.1% of cases. The sensitivity was 80%, the specificity was 93.5%, the positive predictive value was 85.7% and the negative predictive value was 90.62%.

## 4. Discussion

This prospective cohort study followed 52 patients for a minimum of one year, from the moment of RRMS diagnosis. The patient lot’s main demographic and clinical characteristics fell in line with reports from much larger registries regarding sex ratio, average age, etc. [27]. A particularity of this cohort, however, is the large proportion of patients experiencing an active form of the disease from onset, with 19 of them (36.5%) showing a progression of EDSS-plus score after one year of follow-up. This translates to sustained disability accrual and a worse prognosis in the long term [28].

Our leading question was whether easily available biomarkers in use worldwide today may help predict this negative short-term prognosis and aid in the immediate decision-tree following RRMS diagnosis.

The EDSS-plus score, as previously mentioned, was developed to help in detecting the progression of disability in MS and thus was thought out as a useful tool for diagnosing the secondary-progressive phases of the disease [25]. By incorporating the 9-hole peg test and the time to walk 25-feet test, it overcomes some of the main shortcomings of the EDSS score, namely assessing the inability of the upper limb and reliance on a frequently under-investigated ambulatory capacity [29]. The EDSS-plus scale has the advantage of being validated and easy to use, and that it can reliably detect disability progression.

Recent research has brought to light the terms of smouldering MS and progression independent of relapse activity (PIRA), with data from large clinical trials showing that progression may occur early in the evolution of RRMS, and that disability for these patients may be driven mostly by the chronic neurodegenerative processes rather than acute focal inflammatory/demyelinating events [1,30,31]. Our group believes that the EDSS-plus scale better reflects this new understanding of the mechanisms driving disability in MS, making it a useful tool from the very moment of RRMS diagnosis. By analyzing the neurofilament (NfL) levels in our groups, we saw a clearly emerging pattern: initial samples (baseline and 3-month follow-up values) show a trend towards higher adjusted z-scores in the EDSS-plus progressor group (2.01 vs. 1.77 and 1.57 vs. 1.35, respectively), but neither, however, reach statistical significance. The 6- and 12-month follow-up samples show a much clearer difference between the two groups, reaching statistical significance with 1.44 vs. 0.71 (*p* = 0.028) at 6-month follow-up and values of 1.32 vs. 0.49 (*p* = 0.01) at 12-month follow-up.

Serum neurofilaments are elevated several months ahead of clinical events or the progression of disability in MS, serving as a good predictive biomarker. They are also a good biomarker for monitoring treatment response, as patients under high-efficacy treatments or for whom the disease activity is under control show a trend towards a sustained normalization of the sNfL values [7,32].

For our study, EDSS-plus non-progressors show a clear trend towards the normalization of the sNfL values over repeated measurements. EDSS-plus progressors, however, remain at pathologically elevated values over a longer period, correlating with their poorer clinical short-term prognosis. Interestingly, the DMT-type showed no significant influence on the sNFL follow-up samples (3-, 6- and 12-month follow-ups). This may be due to a short time under DMT (median 11 months for this cohort) as well as the very large proportion of patients that were included in moderate-efficacy treatments.

Regarding the OCT metrics, descriptive statistics showed a statistically significant difference between the EDSS-plus progressors and non-progressors, for both RNFL average thickness (90 μm vs. 97 μm, *p* = 0.024) and GCL-IPL average thickness (73 μm vs. 80 μm, *p* = 0.003). This may be due to differences in underlying neurodegenerative processes between the two groups, leading to an increased likelihood of disability progression as was observed in this study [13].

Based on these data, we built a predictive model for the likelihood of the EDSS-plus progressor state at one-year follow-up for newly diagnosed RRMS patients using the BREMSO score, OCT metrics (RNFL and GCL-IPL thickness) and baseline and 6-month follow-up adjusted sNfL z-scores. The model was statistically significant, with χ2(4) = 19.542, *p* < 0.0001 and R2 = 0.791. Performance wise, it correctly classified 89.1% of cases, with a sensitivity of 80%, a specificity of 93.5%, a positive predictive value of 85.7% and a negative predictive value of 90.62%.

We also analyzed whether a reliable prediction model could be built using only variables available at baseline evaluation. After removing the 6-month follow-up sNfL-adjusted z-score from the model, it severely diminished its reliability, while still keeping statistically significant predictive power. We consider that the 6-month follow-up sNfL-adjusted z-scores bring a significant improvement to the prediction model and justify the delay.

We opted against using the RoAD score in our final analysis for two reasons, namely that the RoAD score is only available after one year of follow-up, therefore cancelling any predictive value of this score for this study, and secondly that the score includes the EDSS progression as a parameter, therefore overlapping the notion of the EDSS-plus progressor and explaining the statistically significant difference between the two groups. Long-term follow-up will be of interest regarding how well initial disability progression and long-term prognosis will correlate with this disability prediction tool.

Other author groups have previously observed the increased rates of brain and retinal neuro-axonal damage in the early stages of MS. In this study by Irene Pulido-Valdeolivas et al. [33], patients with active MS showed twice the speed of GCIPL thinning and whole brain volume loss, as well as a thalamic volume loss being five times higher than that of stable MS patients during the first 2 years of follow-up. Age might also be a significant factor, as shown by Cordano et al. [34], with younger patients experiencing faster rates of both retinal and cortical atrophy.

In line with these observations, our study comes to underline that the initial stages of MS may hide dramatic immune-mediated inflammatory activity as well as neurodegenerative processes, leading to an early accrual of disability. While age showed poor correlations with our analyzed metrics, no longitudinal follow-up for OCT metrics was performed within our study and therefore no conclusion can be drawn regarding retinal atrophy rates.

## 5. Conclusions

For newly diagnosed RRMS patients, many centers worldwide still practice escalation therapy, withholding high-efficacy options for aggressive forms of the disease or if the initial moderate-efficacy DMTs are not able to offer adequate control. One of the greatest challenges in the escalation strategy is estimating the short-term prognosis for newly diagnosed RRMS patients.

Our study shows that easily available serum and imaging biomarkers may aid clinicians in this challenge. Our results show that baseline and 6-month follow-up sNfL-adjusted z-scores, RNFL and GCL-IPL average thickness and the BREMSO score can be used to produce a reliable prediction model for the likelihood of EDSS-plus score progressor status at one-year follow-up.

Previous studies have proven the predictive power of both sNfL and OCT metrics (especially for the RNFL) as independent factors correlated with the risk of worsening disability in MS [16,17,35]. Most studies, however, analyzed medium and long-term disability risk, as well as setting cut-off values as determined by their specific cohorts. By combining multiple metrics into a predictive model, we believe this has led to better predictive power and more easy translation of our predictive tool to other cohorts as well.

Our study has a number of drawbacks, such as a small sample size (52 patients), a monocentric design and a short follow-up duration at the time of publishing this article. We also consider another shortcoming to be the need to wait for the 6-month follow-up sNfL sample before the model may be applied, leading to a predictive power of only a very short-term prognosis of 6 months.

This study, however, underlines the importance of implementing predictive biomarkers in daily practice for all MS centers. With OCT examinations and sNfL analysis readily available for most healthcare systems worldwide, we believe such predictive models may help clinicians to better estimate the short-term prognosis of their patients and have a broader picture regarding impending risks at the moment of RRMS diagnosis.

## Figures and Tables

**Figure 1 biomedicines-11-00606-f001:**
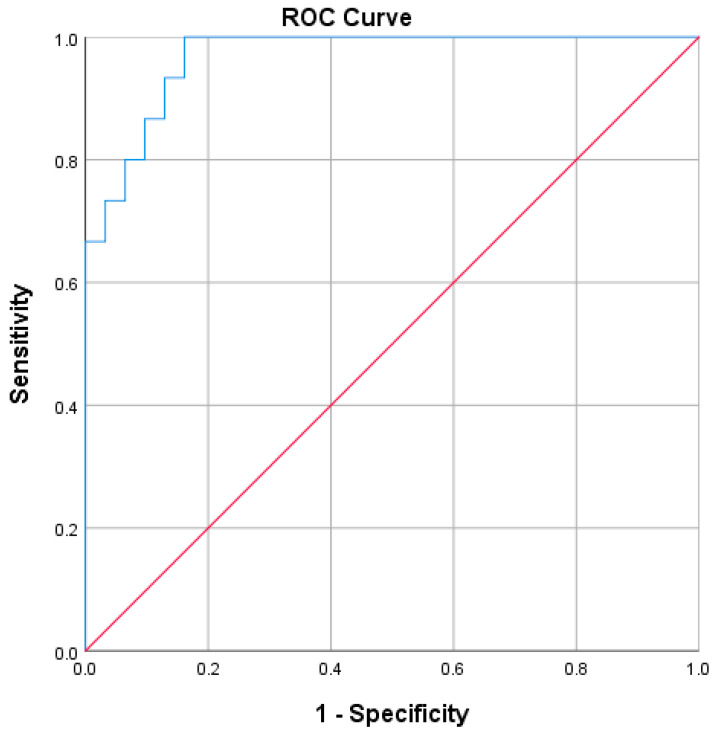
ROC curve of the predictive model for EDSS-plus score progression in the first year following diagnosis of RRMS.

**Table 1 biomedicines-11-00606-t001:** Cohort demographics and main characteristics.

	Median (Minimum and Maximum Values)
Age	29 (18; 52)
Sex	37 F (71.2%)/15 M (28.8%)
Smoker status (active/non-smoker)	15 (28.8%)/37 (71.2%)
Lifestyle (active/sedentary)	37 (71.2%)/15 (28.8%)
Urban/rural	42 (80.8%)/10 (19.2%)
Clinical data	
EDSS—baseline	2 (0; 6)
EDSS—1-year follow-up	1.5 (0; 6.5)
Type of DMT started after study inclusion High-efficacy DMT Moderate-efficacy DMT	8 (15.4%) 44 (84.6%)
Relapses in the first year (no. of) Zero relapses One relapse Two relapses	32 (61.5%) 18 (34.6%) 2 (3.8%)
CSF immunological analysis	No. of patients (%)
Positive oligoclonal bands	37 (71.2%)
Baseline MRI characteristics	
Twenty or more T2/FLAIR hyperintense lesions	37 (71.2%)
Two or more gadolinium-enhancing lesions (GdE)	12 (23.1%)
1-year follow-up MRI characteristics	
Three or more new T2/FLAIR hyperintense lesions	26 (50%)
One or more new GdE	4 (7.7%)
Baseline OCT characteristics	Mean (minimum and maximum values)
RNFL	95.2 µm (67; 118)
GCL + IPL	79.0 µm (56; 92.5)
Neurofilaments *	
Baseline CSF NfL raw values	1114 pg/mL (201; 4210)
Baseline sNfL-adjusted z-score	2.14 (−1.64; 3.81)
Three-month follow-up sNfL-adjusted z-score	1.34 (−0.99; 3.29)
Six-month follow-up sNfL-adjusted z-score	0.98 (−1.8; 2.95)
Twelve-month follow-up sNfL-adjusted z-score	0.81 (−1.85; 2.65)
CSF Beta-amyloid	650 (280; 1211)
Predictive scores	
BREMSO	0.44 (−0.65; 2,39)
RoAD	3 (0; 7)

BREMSO: Bayesian risk estimate for MS at onset; CSF: corticospinal fluid; DMT: disease-modifying treatment; EDSS: expanded disability status score; GCL + IPL: ganglion cell-inner plexiform layer; GdE: gadolinium-enhancing lesions; MRI: magnetic resonance imaging; MS: multiple sclerosis; OCT: optical coherence tomography; RoAD: risk of ambulatory disability score; RNFL: retinal nerve fiber layer; sNFL: serum neurofilaments.* Neurofilament values are presented after adjusting for outliers as described below.

**Table 2 biomedicines-11-00606-t002:** EDSS-plus progressor status confounder analysis.

	EDSS-Plus Progressor	EDSS-Plus Non-Progressors	*p* Value
Age (mean)	32.1 years	29.5 years	0.32
Active smoker	5 (26.3%)	10 (30.3%)	0.76
Masculine sex	5 (26.3%)	10 (30.3%)	0.76
Rural environment	6 (31.6%)	4 (12.1%)	0.14
Sedentary lifestyle	6 (31.6%)	9 (27.3%)	0.74
Positive OCBs	14 (73.7%)	23 (69.7%)	0.76
EDSS baseline score (mean)	2.1	2.0	0.85
Baseline MRI Twenty or more T2/FLAIR hyperintense lesions	15 (78.9%)	22 (66.7%)	0.34
Two or more GdE lesions	6 (31.6%)	76 (18.2%)	0.31
One-year follow-up MRI Three or more new T2/FLAIR hyperintense lesions	10 (52.6%)	16 (48.5%)	0.77
One or more new GdE	1 (1.9%)	3 (5.8%)	1.0
Moderate efficacy DMT type	18 (94.7%)	26 (78.8%)	0.232
Relapses during the first year (no. of) 0 1 2	9 (47.4%) 9 (47.4%) 1 (5.3%)	23 (69.7%) 9 (27.3%) 1 (1.9%)	0.28

DMT: disease-modifying treatment; EDSS: expanded disability status scale; FLAIR: fluid-attenuated inversion recovery; GdE: gadolinium-enhancing lesions; OCBs: oligoclonal bands; MRI: magnetic resonance imaging.

**Table 3 biomedicines-11-00606-t003:** Biomarkers and clinical prediction score differences between the EDSS-plus progressors and non-progressors.

	EDSS-Plus Progressor	EDSS-Plus Non-Progressors	*p* Value
BREMSO	0.56	0.42	0.583
Baseline RoAD score	2.37	2.48	0.752
**RoAD final score**	**4.05**	**3.0**	**0.023**
Baseline sNfL-adjusted z-score	2.01	1.77	0.474
Three-month sNfL-adjusted z-score	1.57	1.35	0.502
**Six-month sNfL-adjusted z-score**	**1.44**	**0.71**	**0.028**
**Twelve-month sNfL-adjusted z-score**	**1.32**	**0.49**	**0.010**
Baseline CSF NfL (pg/mL)	1339	1319	0.946
CSF Aβ42 (pg/mL)	667	685	0.769
**OCT RNFL mean thickness**	**90 µm**	**97 µm**	**0.024**
**OCT GCL-IPL mean thickness**	**73 µm**	**80 µm**	**0.003**

Aβ42: amyloid Beta 42 fraction; BREMSO: Bayesian risk estimate for MS at onset; CSF: ce-rebrospinal fluid; GCL-IPL: ganglion cell-inner plexiform layer; OCT: optical coherence tomograph; RNFL: retinal nerve fiber layer; NfL: neurofilaments; sNfL: serum neurofilaments.

**Table 4 biomedicines-11-00606-t004:** Logistic regression predicting likelihood of EDSS-plus progressor state based on baseline and 6-month follow-up adjusted z-score, OCT parameters (RNFL and GCL-IPL) and BREMSO score.

	*B*	*SE*	Wald	*df*	p	Odds Ratio	95% CI for OR	
							Lower	Upper
BREMSO	−0.801	0.830	0.929	1	0.335	0.449	0.088	2.287
Baseline sNfL-adjusted z-score	−1.681	0.911	3.409	1	0.065	0.186	0.031	1.109
Six-month sNfL-adjusted z-score	3.983	1.560	6.517	1	0.011	53.66	2.52	1141.78
RNFL mean thickness	−0.172	0.109	2.482	1	0.115	0.842	0.679	1.043
GCL-IPL mean thickness	−0.187	0.103	3.274	1	0.70	0.830	0.678	1.043
Constant	27.624	11.525	5.745	1	0.17			

## Data Availability

Collected data were used to produce a pseudonymized dataset, available upon reasonable request from the corresponding author.

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
