# Peer review of "Serum Neurofilaments and OCT Metrics Predict EDSS-Plus Score Progression in Early Relapse-Remitting Multiple Sclerosis"

_biomedicines, 2023, doi:10.3390/biomedicines11020606_

Round 1

Reviewer 1 Report

This work attempts to combine serum biomarkers with OCT metrics and the clinical score BREMSO to predict early disability progression. 

The study is well conducted, and the manuscript was well written. The study's limits are stated correctly: small sample size and short follow-up duration. 

Introduction: I suggest improving the background regarding OCT. Previous publications studied OCT metrics and their ability alone to predict the accumulation of disability. (Martinez-Lapiscina EH, et al. Lancet Neurol. 2016 May;15(6):574-84. doi: 10.1016/S1474-4422(16)00068-5: Cordano C, et al. Neurol Neuroimmunol Neuroinflamm. 2018 Dec 21;6(2):e533. doi: 10.1212/NXI.0000000000000533. PMID: 30697584; PMCID: PMC6340330). Why is the suggested approach potentially better?

Methods: Have the authors considered using the INL as a possible predictor? Recent literature (Sotirchos ES, et al. Progressive Multiple Sclerosis Is Associated with Faster and Specific Retinal Layer Atrophy. Ann Neurol. 2020;87(6):885–896. doi:10.1002/ ana.25738; Cordano C, et al.; Retinal INL Thickness in Multiple Sclerosis: A Mere Marker of Neurodegeneration? Ann Neurol. 2021 Jan;89(1):192-193. doi: 10.1002/ana.25933; Nij Bijvank JA, et al. Interpretation of Longitudinal Changes of the Inner Nuclear Layer in MS. Ann Neurol. 2022 Jul;92(1):154-155. doi: 10.1002/ana.26365.) has described the INL as a possible measure of neurodegeneration related to inflammation. It would be interesting to add this metric and evaluate if the predictive model for EDSS-plus score progression can be improved this way.

Discussion: Recent work has described faster atrophy within the first years of disease duration and a correlation between young age and faster retinal atrophy. (Perez-Miralles F, et al. Clinical impact of early brain atrophy in clinically isolated syndromes. Mult Scler Houndmills Basingstoke Engl. 2013; 19(14):1878-1886. doi:10.1177/1352458513488231; Pulido-Valdeolivas I, et al. Retinal and brain damage during multiple sclerosis course: inflammatory activity is a key factor in the first 5 years. Sci Rep. 2020;10(1):13333. doi:10.1038/s41598-020-70255-z; Cordano C, et al. Neurology. 2022 Oct 11;99(15):e1685-e1693. doi: 10.1212/WNL.0000000000200977). Please comment on the importance of the described findings in light of this literature.

Author Response

To our esteemed reviewer,

We thank you for your feedback. Regarding your comments on this manuscript:

1-    I suggest improving the background regarding OCT. Previous publications studied OCT metrics and their ability alone to predict the accumulation of disability.

The recommended citations were added to the manuscript. This paragraph was added to the text as well:

Previous studies have proven the predictive power of both sNfL and OCT metrics (especially for the RNFL) as independent factors correlated with the risk of worsening disability in MS.[16],[17],[33] Most studies however analysed medium and long-term disability risk, as well as setting cut-off values as determined by their specific cohorts. By combining multiple metrics into a predictive model, we believe this has led to a better predictive power and more easily translation of our predictive tool to other cohorts as well.

2-    Methods: Have the authors considered using the INL as a possible predictor? Recent literature has described the INL as a possible measure of neurodegeneration related to inflammation. It would be interesting to add this metric and evaluate if the predictive model for EDSS-plus score progression can be improved this way.

Thank you for this suggestion. Unfortunately, our colleagues from the Ophthalmology Department were unable to provide us with this metric for our cohort.

3-    Discussion: Recent work has described faster atrophy within the first years of disease duration and a correlation between young age and faster retinal atrophy.  Please comment on the importance of the described findings in light of this literature.

We have added the following paragraphs within the manuscript:

Other author groups have previously observed the increased rates of brain and retinal neuro-axonal damage in the early stages of MS. In this study by Irene Pulido-Valdeolivas et. al[33], patients with active MS showed twice the speed of GCIPL thinning and whole brain volume loss, as well as a thalamic volume loss 5 times higher than that of stable MS patients during the first 2 years of follow-up. Age might also be a significant factor, as shown by Cordano et. al[34], with younger patients experiencing faster rates of both retinal and cortical atrophy.

In line with these observations, our study comes to underline that the initial stages of MS may hide dramatic immune-mediated inflammatory activity as well as neuro-degenerative processes, leading to an early accrual of disability. While age showed poor correlations with our analyzed metrics, no longitudinal follow-up for OCT metrics was performed within our study and therefore no conclusion can be drawn regarding retinal atrophy rates.

We hope that our answers have properly addressed your observations. Please let us know of any other issues with the manuscript.

Kind regards,

Prof. Bogdan. O. Popescu

On behalf of the authors group

Reviewer 2 Report

Identifying disease progression and follow up is one of the main unmet needs for MS researches today. Many biomarkers have been studies and are very promising but still for most of them , results are usually controversial.

Combining two or more such biomarkers can increase the sensitivity of the methods.

In this work you very successfully combine: MRI findings, OCT clinical markers and NFL's

1-    NFL’s are shown to be higher during relapse. Was the first blood sample taken during relapse or how long after? Similarly, if follow up samples were raking during any relapse.

2-    Do you have information for events of Optic neuritis in the past for those patients

3-    You mention that NFL changes were similar in patients started high efficacy treatments and those under moderate efficacy group. Can you specify which medications are in each category? This come to controversy with other publications showing that reduction of NFL's is more significant with high efficacy medications (ie Delcoigne et al, Blood neurofilament light levels segregate treatment effects in multiple sclerosis, Neurology 2020 Mar 17;94(11):e1201-e1212. doi: 10.1212/WNL.0000000000009097. Epub 2020 Feb 11.)

Author Response

To our esteemed reviewer,

We thank you for your feedback. Regarding your comments on this manuscript:

1-    NFL’s are shown to be higher during relapse. Was the first blood sample taken during relapse or how long after? Similarly, if follow up samples were raking during any relapse.

-All initial serum samples were obtained in a 6-months time-frame following the clinical event leading to RRMS diagnosis. The average time frame for baseline serum samples was 3 months since last relapse, with little variation between subjects. We analysed initial samples to see the influence this time frame might have, but found out this was mostly correlated to relapse severity and MRI features rather than the proximity to the relapse. We considered this to be a natural clinical variation that didn’t require further correction.

-Future samples were recruited at regular intervals for all subjects. Acquirement of further disability, whether by new relapses, silent lesions or functional tests worsening was considered to be an endpoint and not a variable/confounder, so we believe the 3, 6 and 12 months samples should not adjust for proximity to relapses.

2-    Do you have information for events of Optic neuritis in the past for those patients

Yes, as mentioned in the manuscript, all eyes with a history of optic neuritis were excluded from final analysis.

3-    You mention that NFL changes were similar in patients started high efficacy treatments and those under moderate efficacy group. Can you specify which medications are in each category? This come to controversy with other publications showing that reduction of NFL's is more significant with high efficacy medications (ie Delcoigne et al, Blood neurofilament light levels segregate treatment effects in multiple sclerosis, Neurology 2020 Mar 17;94(11):e1201-e1212. doi: 10.1212/WNL.0000000000009097. Epub 2020 Feb 11.)

High efficacy treatment options varied between Natalizumab, Ocrelizumab and Fingolimod.

Patients were not on any DMT on baseline samples collection, and for the 3 and 6 months samples, the only samples included in the final analysis, the median time under DMT for the lot was just a little under 2, respectively 5 months. This time frame is too short to observe significant differences between the 2 treatment options, especially for a moderately sized lot of patients. There was a trend towards lower values for the adjusted z-scores at 3, 6 and 12 months for HE DMT patients, but none reached statistical significance.

In conclusion, just as we stated in the manuscript, we believe this to reflect only the short follow-up and a consequence of the low number of patients rather than contradicting existing literature on the subject.

Here is the paragraph from the manuscript that addresses this issue:

For our study, EDSS-plus non-progressors show a clear trend towards the normalization of the sNfL values over repeated measurements. EDSS-plus progressors however remain at pathologically elevated values over a longer period, correlating with their poorer clinical short-term prognosis. Interestingly, DMT-type showed no significant influence on the sNFL follow-up samples (3-,6- and 12- months follow-up). This may be due to a short time under DMT (median 11 months for this cohort) as well as the very large proportion of patients that were included on moderate-efficacy treatments.

We hope that our answers have properly addressed your observations. Please let us know of any other issues with the manuscript.

Kind regards,

Prof. Bogdan. O. Popescu

On behalf of the authors group